# Machine-Based Resistance Training Improves Functional Capacity in Older Adults: A Systematic Review and Meta-Analysis

**DOI:** 10.3390/jfmk9040239

**Published:** 2024-11-16

**Authors:** Archie Kirk, James Steele, James P. Fisher

**Affiliations:** Department of Sport and Health, Solent University, Southampton SO14 0YN, UK; archiemkirk@virginmedia.com (A.K.); james@steele-research.com (J.S.)

**Keywords:** generality of strength, specificity, transfer, activities of daily living

## Abstract

**Background/Objectives**: Resistance training (RT) can improve the functional performance of older adults, maintaining independence and quality of life. It has been proposed that training interventions should implement exercises associated with the movements needed in everyday life. However, this strength training philosophy presents challenges, specifically to older adults, and the use of resistance machines might present an efficacious alternative. The aim of this systematic review and meta-analysis was to explore the impact of machine-based RT on strength and functional capacity in older adults. **Methods**: The inclusion criteria were for strength training interventions to be a minimum of 6 weeks, using only resistance machines, with pre- and post-intervention measurements of functional capacity of either a timed up-and-go and/or a sit-to-stand test, and including healthy older adults (>60 years). **Results**: Following the screening, 17 articles met the inclusion criteria for the systematic review, 15 of which were included in the meta-analysis for functional outcomes (n = 614 participants), and 11 of which were included in the meta-analysis for strength outcomes (n = 511 participants). Analyses revealed significant standardized mean change in favor of machine-based RT for functional outcomes (0.72, 95% CIs 0.39 to 1.07) and strength outcomes (0.71, 95% CIs 0.34 to 1.08) compared to control conditions (functional = 0.09, 95% CIs − 0.1 to 0.28, strength = 0.1, 95% CIs − 0.05 to 0.24). Substantial heterogeneity was noted in the manipulation of RT variables and the magnitude of effects between studies. **Conclusions**: The data presented support the idea that significant strength and functional performance outcomes are attainable using uncomplicated, machine-based RT.

## 1. Introduction

Physical decline with age includes dynapenia and sarcopenia (loss of strength and muscle mass, respectively [1]), as well as decreases in bone mineral density [2]. The loss of strength and muscle mass can contribute to diminished balance, increased risk of falls, and in combination with reductions in bone density, a greater likelihood of fractures and reduced healing capacity [3]. Historically, strength and muscle mass have been independent predictors of longevity [4,5]. However, and perhaps more importantly, the associated function likely serves to maintain independence and quality of life, preventing or delaying institutionalization or hospitalization.

Resistance training is well-documented to support reductions in blood pressure [6], increases in bone mineral density [7], improvements in metabolic health markers [8], and improved physical function in older adults [9,10]. However, it has been proposed that any intervention should implement resistance and balance exercises associated with movements needed in older adults’ everyday lives [11]. In this sense, training specificity might include complex movement patterns that replicate the direction of force, as well as providing instability through multi-planar movements, or at the least multi-planar capacity (e.g., the use of free weights, bands, cables, or bodyweight, where stabilizing a resistance is required) to improve balance, mobility, strength, and activities of daily living [12]. Indeed, a recent meta-analysis [13] supported the notion of specificity within adaptation, that strength increases showed larger Cohen’s dz values in strength exercises being trained. However, the authors also report some generality of strength adaptation, that an individual can make strength and/or functional improvements in movements or exercises that were not trained but recruit similar musculature [14].

While the benefits of functional training are recognized [12], these methods are not without limitations. For example, the functional training methods discussed within the literature, including the use of bands, bodyweight, and weight cuffs, combined with instability might limit the extent of muscular overload (due in part to the instability itself) and be difficult to quantify and progress over the duration of a training program [15,16]. Furthermore, free weights are documented to have a greater injury risk (as well as severity of injury) compared to training using resistance machines [17]. Epidemiological research considering emergency department visits following weight training injuries reported that 90% of injuries were caused by free-weight training, and 65% of injuries were the result of weights dropping on a person [17]. Of course, it stands to reason that, during free-weight exercise, the potential to fall or drop a weight exists and, thus, is a higher risk than a resistance machine, where a person is typically seated and a weight stack moves up and down but never over the user.

In addition to being safer, reviews have supported the use of uncomplicated machine-based resistance training for increases in strength and muscle mass [18,19] as well as health benefits [20]. Finally, and since functional training methods might add a degree of difficulty (e.g., training technique, program design, exercise selection, load progression, force direction, and strength curve), the use of resistance machines might also serve to overcome perceived complexity—an often-cited barrier to resistance training adherence from [21]. However, we should clarify that the use of resistance machines and free weights are not mutually exclusive; training programs might efficaciously choose to incorporate both resistance methods.

Based on the principle of specificity, we recognize that the use of resistance machines, limited in the plane of motion, can train only the movement and the muscles involved in each specific exercise, and thus, any transference to functional capacity outcomes is representative of a generality of strength (e.g., an ability to use muscles trained in one exercise in a separate and seemingly unrelated task). With this in mind, the primary aim of this systematic review and meta-analysis was to explore the impact of machine-based resistance training on strength and functional capacity in older adults. The secondary aim was to review the training program variables (e.g., study duration, frequency, volume, exercise selection, load, and intensity of effort) in relation to the outcome variables (e.g., functional capacity tests).

## 2. Materials and Methods

This systematic review was performed commensurate with Preferred Reporting Items for Systematic Reviews and Meta-Analyses (PRISMA) guidelines [22]. To locate the relevant studies, we comprehensively searched PubMed/MEDLINE using the following Boolean search syntax: (“resistance training” OR “strength training” OR “weight training”) AND (“timed up and go” OR “sit to stand” OR “functional” OR “functional capacity” OR “functional outcomes”). In addition, we screened the reference lists of the articles retrieved and the applicable review papers to uncover any additional studies that might meet the inclusion criteria. The search process was carried out separately by two researchers (JPF and AK). The initial search consisted of screening all titles for duplication, followed by abstracts for studies potentially meeting the inclusion/exclusion criteria. For papers deemed potentially relevant, full texts were evaluated and decisions were then made as to whether a given study warranted inclusion. The search was finalized in February of 2024.

### 2.1. Inclusion/Exclusion Criteria

The studies that were sourced and went through the screening process had to meet the following inclusion criteria: (1) the article must be published in full-text English, (2) the age of participants had to be older than 60 years, (3) the population sampled was characterized as being healthy or asymptomatic (the decision to choose only apparently healthy adults eliminated the risk of selection bias, as the results for clinical participants might not be representative of typical adaptations or given the arm-based approach used for meta-analysis might be more likely to be influenced by regression to the mean and natural history effects), (4) the study design had to have a pre- and post-test measurement for functional capacity by including either a timed up-and-go and/or a sit-to-stand test, (5) the training intervention had to be at least 6 weeks in duration, and (6) the training intervention could not include free-weight or other (e.g., dumbbells, barbells, kettlebells, sandbags, resistance bands, etc.) resistance types that might represent functionally similar/specific exercise [23]. The exclusion criteria included anything contradictory to the inclusion criteria, in addition to any patients reported as previously suffering from long-term medical conditions that might impede functional performance or inhibit recovery (including but not limited to cancer, stroke, heart disease, diabetes, hypertension, Parkinson’s, and osteoporosis), and/or previous/current musculoskeletal injuries (for example, knee replacement, hip replacement, ACL reconstruction, etc.). Finally, any degree of concurrent/additional training, which included aerobic exercise, balance exercises, stretching, calisthenics, or others was excluded, since these additional training modalities might confound the estimation of the effects of machine-based resistance training alone.

Specifically, we chose the two measures of functional capacity, namely timed up and go and sit to stand, because the acceptance and simplicity of the protocols result in the frequency of their use in assessment and their relationship to other functional measures. The timed up and go requires a participant to move from a seated position to stand and walk to/around an object and return to their seat [24]. This test is used to assess walking speed/ability, dynamic balance, fall risk, and agility, and is correlated to the Berg balance score (r = 0.81), gait speed (r = 0.60), and the Barthel index of activities of daily living (ADL’s; r = 0.78) [25,26]. The sit-to-stand test is typically performed as either the time to perform 5 movements from seated to standing upright or the total number of vertical stands from a seated position within a 30 s time (both performed with the hands folded across the chest to prevent the use of upper body strength [27]). The sit-to-stand test has high interrater reliability (ICC = 0.94; [28]) and correlates well with other functional measures, such as stair climb speed, walking speed (r = 0.52), dynamic balance (r = 0.65), and risk of falling [29,30]. It is also worth noting that the automation and objective quantifications of these clinical assessments appear to be relatively simple with the use of an accelerometer found within a smartphone [24].

### 2.2. Data Extraction

Where studies included multiple training conditions, only the intervention groups that met the identified criteria and any respective non-training control groups were included in the review. Data were extracted for participant characteristics (age), intervention duration, training frequency, training volume, training load, repetition duration, and intensity of effort (Table 1), and for resistance training exercises and resistance type (Table 2). Further data were extracted for functional capacity outcomes (e.g., sit to stand and timed up and go) and any strength outcomes from these studies, including one repetition maximum tests or isometric maximal voluntary contractions. Where only figures were presented, the data were acquired using online software (juicr package; [31]).

### 2.3. Methodological Quality

The methodological quality of each study was evaluated using the Standard Method for Assessment of Resistance Training in Longitudinal Designs (SMART-LD), which has been used to validate the methodological quality of randomized control trials involving resistance training interventions with acceptable inter-rater reliability. This scale has been specifically developed for studies looking at resistance training interventions and appears to be more applicable than using the physiotherapy evidence-based scale (PEDro scale) [47]. The SMART-LD includes 20 criteria across 5 categories, with qualitative methodological ratings for assessment of: “good quality” (16–20), “fair quality” (12–15), and “poor quality” (0–11).

### 2.4. Statistical Analysis

A statistical analysis of the data extracted was performed in R (v 4.3.3; R Core Team, https://www.r-project.org/ accessed 22 April 2024) and RStudio (v 2023.06.1; Posit, https://posit.co/ accessed 22 April 2024). All of the code utilized for the data preparation and analyses are available on either the Open Science Framework page for this project https://osf.io/5fjq3/ (accessed 22 April 2024) or the corresponding GitHub repository at https://github.com/jamessteeleii/older_adults_RT_machines_MA (accessed 22 April 2024). The present analysis was not pre-registered, as we had no a priori hypotheses and, thus, given the nature of this study, was considered exploratory and aimed at parameter estimation [48] within a Bayesian meta-analytic framework [49]. However, at the request of the editor, we have retrospectively registered the study at https://osf.io/e4586 (accessed 22 April 2024), although we note the process of retrospective registration offers value only in so far as the work can be identified in searches of this registry. For all analyses, the model parameter estimates and their precision, along with conclusions based upon them, were interpreted continuously and probabilistically, considering the data quality, the plausibility of effect, and the previous literature, all within the context of each model. The renv package [50] was used for package version reproducibility and a function-based analysis pipeline using the targets package [51] was employed (the analysis pipeline can be viewed by downloading the R Project and running the function targets::tar_visnetwork()). Standardized effect sizes were all calculated using the metafor package [52]. The main package brms [53] was used in fitting all the Bayesian meta-analysis models. Prior and posterior draws were taken using tidybayes [54] and marginal effects [55] packages. All visualizations were created using ggplot2 [56], tidybayes, and the patchwork [57] packages. Where the data to be extracted from included studies were reported in plots only, we used the juicr package to extract this data [31], and the reproducible reports for this can be found at https://osf.io/e4586.

As noted, we adopted a Bayesian approach to the present meta-analysis. Specifically, we adopted an arm-based multiple treatment comparison (i.e., network)-type model, given that, for the studies identified, some, but not all, included a non-training control arm in addition to the machine-based RT arm [58]. In typical contrast-based meta-analyses, data are limited to the effect sizes for paired contrasts between arms and, thus, studies that include both arms (i.e., relative effects between non-training control vs. machine-based RT conditions). However, in arm-based analyses, the data are the absolute effects within each arm, and the information is borrowed across studies to enable both within-condition absolute and between-condition relative contrasts to be estimated. As in the present analysis, we are only comparing two conditions. We do not examine ranking methods, as are typical in multiple treatment comparison models, but instead, we focus on reporting the between-condition relative contrast for non-training control vs. machine-based RT. We fit two models, with one for all of the function outcomes reported and one for all of the strength outcomes reported. Pre- to post-intervention period standardized effect sizes were calculated using Becker’s d [59] for each outcome within each arm within each study. As such, the data were hierarchical across three levels (i.e., effects within arms within studies), and so, we included random intercepts using implicit nested coding across these levels. Given the arm-based model, we included a fixed categorical predictor using dummy coding, indicating which condition a given arm within the study belonged to (i.e., non-training control or machine-based RT, where the former was the intercept) and also allowed for this to be a random effect to enable the partial pooling of information across studies where there were no direct between-condition relative contrasts present. We did not have any prior intuition or data available for the specific intervention in this population that was not included in the likelihood for the model anyway, and so, we adopted uninformed default weakly regularizing priors for all parameters. We fit each model using four Monte Carlo Markov chains, each with 2000 warmup and 6000 sampling iterations. Trace plots were produced along with R^^^values to examine whether the chains had converged, and posterior predictive checks for each model were also examined to understand the model’s implied distributions. These all showed good convergence with all R^^^values close to 1, and posterior predictive checks seemed appropriate distributions for the observed data (all diagnostic plots can be seen at: https://osf.io/z9u7s (accessed 22 April 2024). From each model, we obtained draws from the posterior distributions for the conditional absolute estimates for each condition by study, the global grand mean absolute estimates for each condition, and the between-condition relative contrast for the non-training control vs. machine-based RT in order to present probability density functions visually and also to calculate the mean and 95% quantile intervals (i.e., ‘credible’ or ‘compatibility’ intervals) for each estimate. These gave us the most probable value of the parameter, in addition to the range from the 2.5% to 97.5% percentiles.

## 3. Results

### 3.1. Search Results

For this review, a total of 3249 studies were identified through database searching only. Once duplicates had been removed, 1252 papers were screened for potential eligibility through title screening. This process resulted in the removal of a further 1115 papers, consequently leaving 137 research articles. Following the abstract screening, 54 papers were excluded, leaving 83 research articles. Following the full-text screening of research interventions and participant details, 15 articles met the inclusion criteria stated above. Two additional articles were found after searching the reference lists, bringing the total to 17 articles included in this systematic review [16,27,32,33,34,35,36,37,38,39,40,41,42,43,44,45,46]. The screening process is outlined in Figure 1.

### 3.2. Quality Assessment

Using the SMART-LD scale, eight studies were considered to be of good quality (values of 16–20/20) and nine were considered to be of fair quality (values of 12–15/20). The study quality was assessed by JPF and AK, and the outcome for SMART-LD is presented in Table 3.

### 3.3. Participant Characteristics

All of the training studies reported the participants as being untrained (not currently, or historically engaging in any strength training intervention). In total, there were 897 participants across the 17 studies, 630 of which were female (70%). Four studies included only female participants [16,32,34,40], while the remaining studies included both male and female participants. The mean age within the studies varied from 63.9 years [39] to 78.9 years [36], and when considered in view of the participant samples from each study, the mean age across the 17 studies was 70.2 years.

### 3.4. Exercise Selection and Resistance Type

Of the 17 studies, 6 used pneumatic resistance machines [16,27,33,35,37,43], one used a plate-loaded resistance machine [27], and 7 studies used selectorized resistance machines [32,36,38,39,42,45,46], or any combination. Finally, four studies did not clarify the resistance type or the manufacturer and model (though they did state the use of machine-based resistance). Three studies considered only lower-body resistance training exercises [40,43,45], and the remainder included both upper- and lower-body exercises. The lower body exercises used with the highest frequency were leg press, leg extension, and leg curl, in 15, 10, and 8 studies, respectively. The upper-body exercises used with the highest frequency were chest press, latissimus dorsi pulldown, and seated row, in 11, 8, and 8 studies, respectively. On average, the studies included 6 different exercises (3 upper body and 3 lower body), with a maximum of 12 different exercises [27] and a minimum of 2 different [43]. See Table 2 for full details of the exercises and resistance types.

### 3.5. Study Duration and Frequency

Of the 17 studies included, the duration ranged from a minimum of 6 weeks [40] up to 12 months [42]. The median study duration was 12 weeks (interquartile range; IQR = 6). Within the studies, the training frequency was either 2 or 3x/week, with a median frequency of 2 (IQR = 2). Accepting that not all participants had 100% attendance in each study, the maximum number of possible workouts per intervention ranged from 12 training sessions [40] to 104 training sessions [42], with a median total training volume of 25.8 training sessions (IQR = 12).

### 3.6. Volume, Effort, Load, and Repetition Duration

Some of the studies utilized simple training programs where the volume of sets and repetitions did not vary throughout the intervention [39]. Some studies incorporated a progression of sets and repetitions as the intervention continued [37], while other studies had multiple training groups that were often prescribed different volumes of sets and repetitions [34]. While Table 1 reports the training volume for each study in detail, the average sets and repetitions in each study were calculated, and from this, the average across all studies is 2.5 ± 0.6 sets of 11.0 ± 3.0 repetitions. Ratings of perception of effort were reported within 11 of 17 studies. However, these were in different formats between studies, e.g., 10-point scale [27,33,34], 6–20-point scale [36,41], or actual effort was proxied by the specific training protocol including the use of a repetition maximum [16,35,37], and concentric failure [32,40,46] to reflect maximal effort. The remaining six studies did not report the effort of the participants in any format [38,39,42,43,44,45]. Table 1 includes all details of the effort measures reported.

Most studies reported training load as a % 1 repetition maximum (RM), or occasionally % other RM (e.g., 10 RM [34]; 8 RM [36]). In many cases, the training load varied throughout the intervention or between training conditions where there were multiple training groups. Four studies did not report a training load [16,27,40,42]. Where studies reported %10 RM or %8 RM, Table 17.8 from NSCA [60] was used to calculate %1 RM. For ease of consideration, the average load was calculated as the mean load across all intervention groups across all studies as = 63 ± 23%1 RM.

The repetition duration was not reported in 8 of the 17 studies [32,36,37,38,40,41,42,45]. Four studies reported concentric muscle actions as being performed as fast as possible [27,33,43,44]. The remaining five studies reported concentric muscle actions as ~2 s, and all nine studies that reported repetition duration stated the eccentric muscle action as ~2 s.

### 3.7. Functional Outcomes

The model examining functional outcomes included 15 studies containing 33 separate arms (8 non-training control and 25 machine-based RT) reporting 54 within arm effects. The global grand mean estimate for the between-condition relative contrast (i.e., machine-based RT minus non-training control) was 0.63 [95% credible interval: 0.23,1.04], though there was considerable heterogeneity in the magnitude of effects between studies (τ_Condition (training) = 0.66 [95% credible interval: 0.37,1.07]). An ordered forest plot of conditional study level estimates for the absolute within-condition effects, the global grand mean estimates for absolute within-condition effects, and the global grand mean estimate for the between-condition relative contrast, including interval estimates and posterior probability distributions, are shown in Figure 2.

### 3.8. Strength Outcomes

The model examining strength outcomes included 11 studies containing 24 separate arms (6 non-training control and 18 machine-based RT) reporting 60 within arm effects. The global grand mean estimate for the between-condition relative contrast (i.e., machine-based RT minus the non-training control) was 0.61 [95% credible interval: 0.21,1.01], though there was considerable heterogeneity in the magnitude of effects between studies (τ_Condition (training) = 0.57 [95% credible interval: 0.32,1.00]). An ordered forest plot of conditional study level estimates for absolute within-condition effects, the global grand mean estimates for absolute within-condition effects, and the global grand mean estimate for the between-condition relative contrast, including interval estimates and posterior probability distributions, are shown in Figure 3.

## 4. Discussion

This represents the first systematic review with a meta-analysis that considers the effects of machine-based resistance training alone upon functional outcomes in older adults. A previous review appears superficially similar [15]. However, the authors included anthropometric changes relating to sarcopenia, as well as studies that had incorporated multi-modal training methods (e.g., cardiovascular, free-weight, and balance training). Instead, we were interested specifically in whether training using a non-specific modality, namely machine-based resistance training, would alone result in improvements in functional outcomes in older adults.

First, our data confirmed significant strength increases as a result of machine-based strength training interventions. Values were of a similar magnitude to those reported previously in both large-scale meta-analyses [61] and in a large sample including older adults [62]. While this might be expected, it should not be overlooked. There is a recognized and well-documented decline in strength in females > 40 years and males > 60 years [63]. Any methods that can mitigate or reverse this decline are likely to be important for prolonged health and longevity. Machine-based resistance training has been presented as a time-efficient and uncomplicated approach to strength training [18,20], and engagement in resistance training, and concomitant increases in strength are associated with reductions in all-cause mortality [64,65].

Second, our analyses show significant improvements in functional outcome measures as a result of machine-based resistance training in older adults. Specifically, we considered two measures of functional capacity; timed up and go and sit to stand. While we recognize alternate methods of measuring functional capacity exist, we elected to include only these two assessments because of their prevalence in the literature, their relationship with other activities of daily living [25,26,29,30], and because of the acceptance and simplicity of the protocols. Previous research has suggested a need for task-specific training [11], and indeed, strength transference is a continued area of research [13]. Based on our analyses, we propose that there is not a need to attempt to perform balance tasks, replicate activities, or recreate superficially similar motor schema (e.g., adding load to a movement). But, increasing muscular strength through uncomplicated machine-based resistance training can then be applied to activities of daily living, or at the very least, a machine-based resistance training intervention alone can, through the totality of its causal mechanisms, improve functional outcomes. We believe these data add to our understanding of training specificity and application of strength and posit that any recommendation to train a movement rather than a muscle might be unnecessary. Our data suggest that a person can train a muscle(s) to be stronger and then apply that strength in seemingly unrelated tasks. Strength transfer and specificity have previously been discussed in the more specific context of strength and conditioning, where evidence supports that motor schema are specific in biomechanics, including load and velocity, and that adding resistance to sporting movements can negatively impact performance [18].

We should recognize that, while this review included 17 empirical research studies, there was considerable disparity in the application of the training variables, including study duration, training volume, load, repetition duration, and effort. However, even the shortest duration intervention (6 weeks; [40]) reported considerable strength and functional outcome increases. While manipulation of these training variables might, in the future, support differing degrees of adaptation, at present, we suggest that the consistently positive outcomes reported herein permit a degree of freedom to all personal trainers or clinical exercise physiologists prescribing exercise to older adults.

### Limitations

We should state that this systematic review and meta-analysis was not pre-registered using Prospero, OSF, or any other. We wish to clarify that this study did not ask the question “do increases in strength from machine-based resistance training causally mediate improvements in functional capacity?”. Thus, our comments above regarding the transference of strength remain speculative. We acknowledge that there might be a disparity between strength increases on trained exercises and functional capacity outcomes. Further, machine-based resistance training might improve functional measures, even in the absence of detectible strength increases. For example, a person engaging in resistance training might experience increases in confidence or perception of health, which manifest as improvements in physical function. However, our analyses present evident and significant strength increases, as well as improvements in functional capacity measures as a result of strength training interventions. Thus, it seems reasonable to speculate that they may be causally related. Further, we should clarify that we are not opposed to alternate training methods that include the use of free weights, bands, or most recently, the use of a weighted vest [66] or the replication of functional tasks, when performed safely. However, our data supports that these are unnecessary at the initiation of a strength training intervention.

## 5. Conclusions

The present systematic review and meta-analysis have shown that uncomplicated, machine-based resistance training can increase strength, as well as functional capacity. Such improvements might serve to preserve independence and improve quality of life. When considering the practical implications of these findings, it is important to recognize the consistently positive outcomes in relation to the heterogeneous manipulation of training variables. We propose that personal trainers or clinicians working with older adults can prescribe a strength training intervention using resistance machines with leniency around other variables without a need to challenge balance or replicate movement patterns. Further research, considering strength transference and specificity of adaptation, should continue to compare strength and performance increases between superficially similar and seemingly unrelated tasks.

## Figures and Tables

**Figure 1 jfmk-09-00239-f001:**
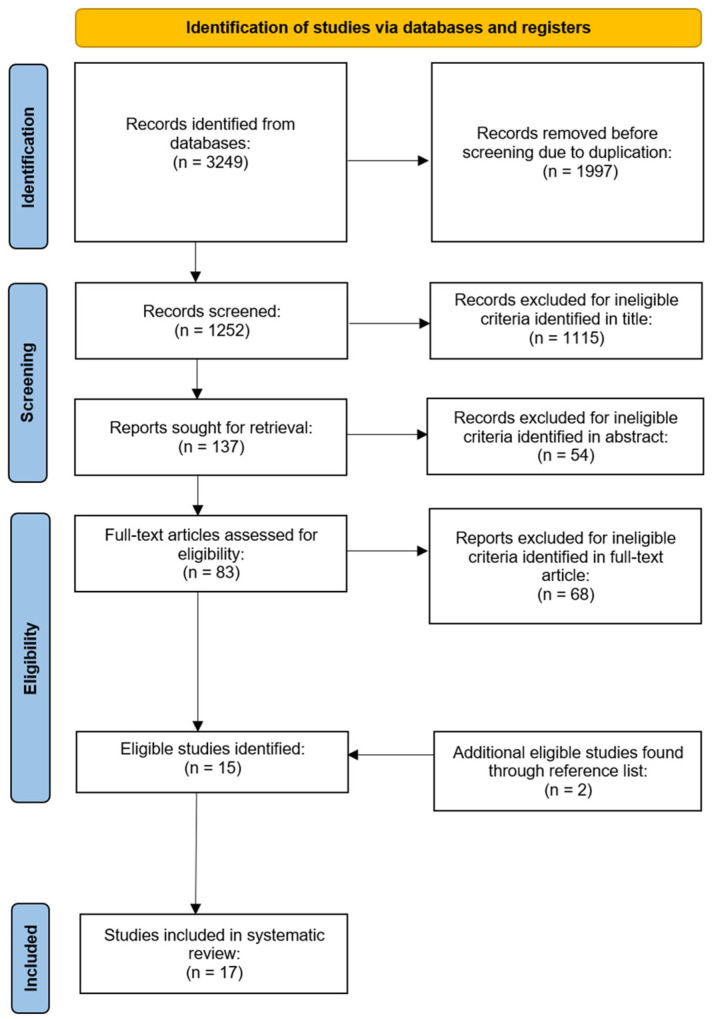
PRISMA flow chart for the screening process.

**Figure 2 jfmk-09-00239-f002:**
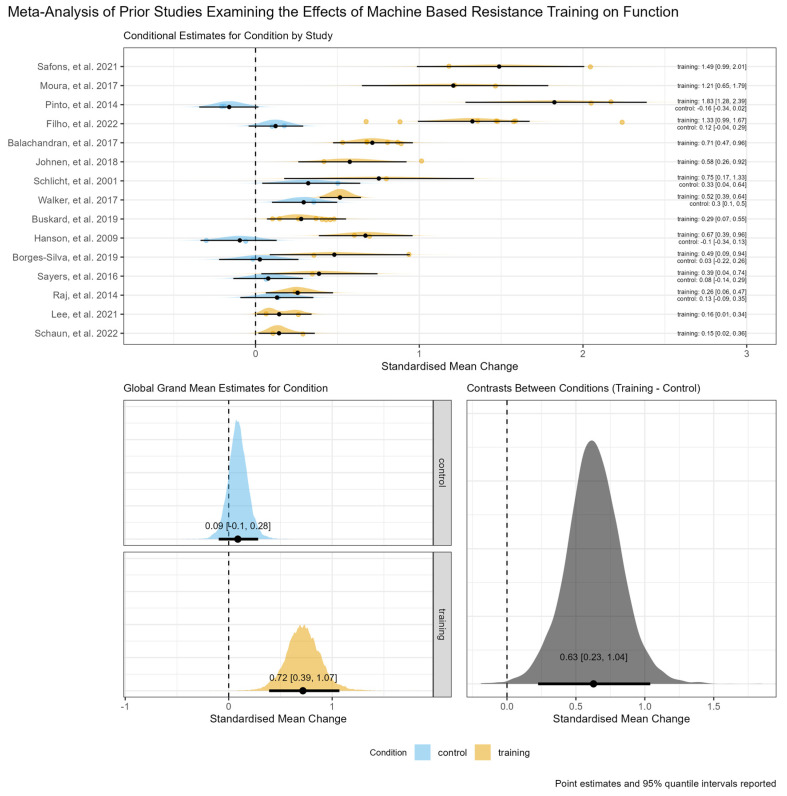
An ordered forest plot for functional outcomes of conditional study level estimates for absolute within-condition effects including individual effect sizes as points (**top panel**), the global grand mean estimates for absolute within-condition effects (**bottom left panel**), and the global grand mean estimate for the between-condition relative contrast (**top right panel**), including interval estimates and posterior probability distributions [16,27,32,33,34,35,36,37,39,40,41,43,44,45,46].

**Figure 3 jfmk-09-00239-f003:**
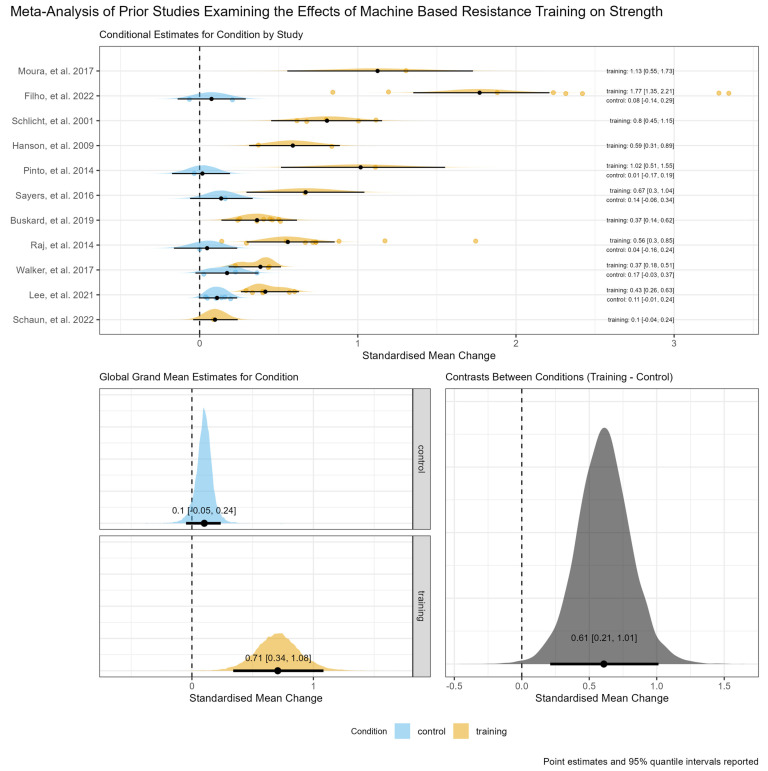
An ordered forest plot for strength outcomes of conditional study level estimates for absolute within-condition effects, including individual effect sizes as points (**top panel**), the global grand mean estimates for absolute within-condition effects (**bottom left panel**), and the global grand mean estimate for the between-condition relative contrast (**top right panel**), including interval estimates and posterior probability distributions [33,34,35,37,39,40,41,43,44,45,46].

**Table 1 jfmk-09-00239-t001:** Resistance training program and variables from each included study. Training groups; RiR = repetitions in reserve, RM = repetition maximum, RPE = rating of perceived exertion, SET = strength–endurance training, PWT = power weight training, AST = absolute strength training, TRT = traditional resistance training, Trad = traditional, ECC = eccentrically biased repetition durations; Conc = concentric, Ecc = eccentric, DNS = did not state.

Authors	Participant Characteristics by Group	Intervention Duration	Training Frequency	Training Volume	Training Load	Repetition Duration	Effort
Balachandran, et al., 2017 [27]	Plate loaded; n = 17 (8 female), 68.8 ± 5.0 years Pneumatic; n = 19 (12 female), 68.9 ± 4.9 years	12 weeks	2x/week	Weeks 1–7: 3 sets of 10 repetitions Weeks 8–12: 3 sets of 8 repetitions	DNS	Conc = as fast as possible, Ecc = 2 s	RPE of 6–8/10
Borges-Silva, et al., 2022 [32]	Traditional Resistance Training; (n = 15 female), 64.2 ± 4.0 years Circuit Resistance training; (n = 15 female), 64.7 ± 4.4 years	12 weeks	2x/week	1–3 sets of 6 RM	85–90% 1 RM	DNS	Failure
Buskard, et al., 2019 [33]	RiR; n = 21 (13 female), 72.3 ± 5.7 years %1 RM; n = 20 (13 female), 69.6 ± 7.4 years RM; n = 21 (13 female), 72.3 ± 6.6 years RPE; n = 20 (13 female), 71.8 ± 6.2 years)	11 weeks	2x/week (weeks 1–5) 3x/week (weeks 6–9)	RiR; 3 sets of 7 repetitions %1 RM; 3 sets @80% 1 RM RM; 3 sets of 8 RM RPE; 3 sets of 7 repetitions	80% 1 RM	Conc = as fast as possible, Ecc = 2 s	RiR group: 1 repetition RM group: failure RPE group: ≤8/10
Filho, et al., 2022 [34]	SET; n = 20 female, 65 ± 4 years PWT; n = 18 female, 66 ± 4 years AST; n = 21 female, 66 ± 5 years TRT; n = 17 female, 67 ± 4 years	20 weeks	2x/week	SET: 1 set of 20–25 repetitions PWT: 2–3 sets of 8–12 repetitions AST: 4–5 sets of 4–5 repetitions TRT: 2–3 sets of 8–12 repetitions	SET, AST, TRT: 60% 10 RM PWT; 50% 10 RM	SET, AST, TRT: Conc = 2 s, Ecc = 2 s PWT: max velocity	RPE of 6–8/10
Hanson, et al., 2009 [35]	n = 50 (27 females); 71.0 ± 5.0 years	22 weeks Phase 1: 10 weeks Phase 2: 12 weeks	3x/week 3x/week	Phase 1: 5 repetitions at 50% of 1 RM Set 2: 5 repetitions at 5 RM Set 3: 10 repetitions at 5 RM * Set 4: 15 repetitions at 5 RM * Set 5: 20 repetitions at 5 RM * Phase 2: 5 repetitions at 50% of 1 RM Set 2: 15 repetitions at 5 RM *	85% 1 RM	Conc = 2 s, Ecc = 3 s	Repetition maximum
Johnen, et al., 2018 [36]	n = 14 (8 female), 78.9 ± 9.11 years	12 weeks	2x/week	1 set of 18–20 repetitions 1 set of 10–12 repetitions	50% of 8 RM 75–80% of 8 RM	DNS	RPE 12/20
Lee, et al., 2021 [37]	n = 234 (174 females) for 12-week analysis, n = 106 (82 females) for 24-week analysis 73 ± 6.5 years	12 and 24 weeks	2x/week	Weeks 1–2: 2 sets of 15 RM Weeks 3–8: 2 sets of 10 RM Weeks 8–12: 2 sets of 8 repetitions Weeks 13–24: 3 sets of 10 repetitions	Weeks 1–2: ~60% 1 RM Weeks 3–7: ~70% 1 RM Weeks 8–12: ~80% 1 RM Weeks 13–24: ~70% 1 RM	DNS	Repetition Maximum
Leenders, et al., 2013 [38]	n = 24 females, 71 ± 1 years n = 29 males, 70 ± 1 years	24 weeks	3x/week	Weeks 1–4: 4 sets (lower body) of 10–15, 3 sets (upper body) of 10–15 Week 5–24: 4 sets of 8 repetitions	Weeks 1–4: 60%–75% 1 RM Week 5–24: 75–80% 1 RM	DNS	DNS
Moura, et al., 2017 [39]	n = 15 (10 females), 63.9 ± 3.0 years	12 weeks	2x/week	3 sets of 10 repetitions	60–90% 1 RM	Conc = 2 s, Ecc = 2 s	DNS
Pinto, et al., 2014 [40]	n = 19 females, 66.0 ± 8 years	6 weeks	2x/week	Weeks 1–3: 2 sets of 15–20 RM Weeks 4–6: 3 sets of 12–15 RM	DNS	DNS	Concentric failure
Raj, et al., 2014 [41]	Trad: n = 12 (5 females), 68 ± 5 years Ecc = 13 (5 females), 68 ± 5 years	16 weeks	2x/week	Trad = 2 sets of 10 repetitions Ecc = 3 sets of 10 bilateral concentric repetitions, and unilateral eccentric repetitions	Traditional = 75% 1 RM Eccentrically biased = 50% 1 RM	DNS	RPE 12–15/20
Roma, et al., 2013 [42]	Resistance training group: n = 20 (17 females), 68.8 ± 5.6 years	12 months	2x/week	3 sets of 12, 10, and 8, repetitions, respectively	DNS	DNS	DNS
Safons, et al., 2021 [16]	n = 23 female, 67.5 ± 5.18	12 weeks	2x/week	Weeks 1–4: 3 sets of 12 repetitions Weeks 5–8: 3 sets of 10 repetitions Weeks 9–12: 3 sets of 8 repetitions	DNS	Conc = 2 s, Ecc = 2 s	Repetitions maximum
Sayers, et al., 2016 [43]	n = 28 (17 female), 71.5 ± 6.8 years	12 weeks	3x/week	3 sets of 14 repetitions	40% 1 RM	Conc = as fast as possible, 1 s pause, Ecc = 2–3 s	DNS
Schaun, et al., 2022 [44]	Older aged adults: n = 18 (9 female), 68.9 ± 6.5 years	12 weeks	2x/week	Week 1: 1 set Weeks 2–6: 2 sets Weeks 7–12: 3 sets All of 8–10 repetitions	1x/week = 40% 1 RM 1x/week = 60% 1 RM	Conc = as fast as possible, Ecc = 2 s	DNS
Schlicht, et al., 2001 [45]	n = 11 (7 female), 72 years	8 weeks	3x/week	2 sets of 10 repetitions	75% of 1 RM	DNS	DNS
Walker, et al., 2017 [46]	n = 46 female, 68.6 ± 2.0 years, n = 35 males, 69.8 ± 2.4 years	12 weeks	2x/week	Weeks 1–4 (initiation): 2 sets of 16–20 repetitions Weeks 5–12 (superset): 2 or 3 sets of 14–16 repetitions	50–60% 1 RM	Conc = 2 s Ecc = 2 s	Volitional concentric failure

* Repetitions begun with 5 RM load and then the load reduced to allow more repetitions, repeated until the desired number of reps completed.

**Table 2 jfmk-09-00239-t002:** Resistance training exercises and resistance type.

Authors	Upper Body Exercises	Lower Body Exercises	Resistance Machine and Manufacturer
Balachandran, et al., 2017 [27]	Chest press, seated row, shoulder press, latissimus dorsi pulldowns, biceps curl, triceps pushdowns	Leg press, leg curl, calf raises, hip abduction, hip adduction	Plate loaded; Cybex VR2, Cybex, Franklin Park, IL, USA or Pneumatic Resistance; Keiser A420, Keiser, Freson, CA, USA
Borges-Silva, et al., 2022 [32]	Pec deck fly, seated row, preacher curl	Prone leg curls, seated calf raises, leg extension	Technogym selectorized, Technogym, Gambettola, Italy
Buskard, et al., 2019 [33]	Chest press, seated row, latissimus dorsi pulldown, biceps curl, triceps press down	Seated leg press, leg curl, hip adduction	Pneumatic Resistance; Keiser A420, Keiser, Freson, USA
Filho, et al., 2022 [34]	Seated row, flexor chair, articulated bench press, curl-ups	Leg press, plantar flexion	Not specified
Hanson, et al., 2009 [35]	Chest press, seated row, abdominal crunch	Knee extension, seated leg curl, alternating leg press	Pneumatic resistance; Keiser A300, Keiser, Freson, USA
Johnen, et al., 2018 [36]	Latissimus dorsi pulldown, elbow and shoulder extension, back extension	Leg press,	Proxomed; Compass, Proxomed, Luhden, Germany
Lee, et al., 2021 [37]	Chest press, latissimus dorsi pulldown, abdomen, back extension	Leg press, leg extension, leg curl, hip abduction, hip adduction	Pneumatic resistance; Gym Tonic, Kokkola, Finland
Leenders, et al., 2013 [38]	Chest press, horizontal row, alternating vertical latissimus dorsi pulldown, abdominal crunches, biceps curl, triceps extension	Leg press, leg extension	Technogym selectorized, Technogym, Gambettola, Italy
Moura, et al., 2017 [39]	Latissimus dorsi pulldown, seated cable row, lumbar extension	45° leg press, hip abductor	Righetto, Freestyle, São Paulo, Brazil
Pinto, et al., 2014 [40]		Leg press, knee extension, knee flexion	Not specified
Raj, et al., 2014 [41]	Smith machine bench press, latissimus dorsi pulldown,	45° leg press, calf press	Not specified
Roma, et al., 2013 [42]	Chest press, sit ups, lower back	Leg press, calf press	Biodelta, maxiflex, Joinville, Brazil
Safons, et al., 2021 [16]	Bench press, high pull, triceps, and row	knee flexion, knee extension, hip abduction, hip adduction, hip extension	Pneumatic Resistance, ENDynamic, Enraf Nonius, Rotterdam, The Netherlands
Sayers, et al., 2016 [43]		Leg press, knee extension	Pneumatic Resistance; Keiser A420, Keiser, Freson, USA
Schaun, et al., 2022 [44]	Chest press, seated row	Leg press, knee extension, seated plantar flexion,	Not specified
Schlicht, et al., 2001 [45]		Leg extension, leg press, calf press, Hip adduction, hip abduction Glute Press	Universal 8-station, Universal, Cedar Rapids, IA, USA Paramount Fitness, Los Angeles, CA, USA Cybex International, Cybex, Owatonna, MN, USA
Walker, et al., 2017 [46]	Chest press, latissimus dorsi pulldown, triceps pushdown, abdominal curl, back extension	Leg press, knee extension, knee flexion,	Precor Vitality Series, Precor Inc., Greater Seattle, WA, USA

**Table 3 jfmk-09-00239-t003:** Standard method for assessment of resistance training in longitudinal designs (SMART-LD).

Criteria	Balachandran, et al. 2017 [27]	Borges-Silva, et al. 2022 [32]	Buskard, et al. 2019 [33]	Filho, et al., 2022 [34]	Hanson, et al. 2009 [35]	Johnen, et al., 2018 [36]	Lee, et al., 2021 [37]	Leenders, et al. 2013 [38]	Moura, et al., 2017 [39]	Pinto, et al., 2014 [40]	Raj, et al., 2014 [41]	Roma, et al., 2013 [42]	Safons, et al., 2021 [16]	Sayers, et al., 2016 [43]	Schaun, et al., 2022 [44]	Schlicht, et al., 2001 [45]	Walker, et al., 2017 [46]
Item 1	Yes	Yes	Yes	Yes	Yes	Yes	Yes	Yes	Yes	Yes	No	Yes	Yes	Yes	Yes	Yes	Yes
Item 2	No	No	No	No	No	No	No	No	No	No	No	No	Yes	No	No	No	No
Item 3	Yes	No	Yes	Yes	No	No	Yes	No	No	Yes	Yes	No	Yes	No	Yes	No	Yes
Item 4	Yes	Yes	Yes	Yes	No	Yes	Yes	Yes	No	Yes	Yes	Yes	Yes	Yes	Yes	Yes	Yes
Item 5	Yes	Yes	Yes	Yes	Yes	Yes	Yes	Yes	Yes	Yes	Yes	Yes	Yes	Yes	No	No	Yes
Item 6	Yes	Yes	Yes	Yes	Yes	Yes	Yes	Yes	Yes	Yes	Yes	Yes	Yes	Yes	Yes	Yes	Yes
Item 7	No	Yes	Yes	No	Yes	No	Yes	Yes	Yes	Yes	No	No	No	Yes	Yes	Yes	Yes
Item 8	Yes	Yes	Yes	Yes	Yes	Yes	Yes	Yes	Yes	Yes	Yes	Yes	Yes	Yes	Yes	Yes	Yes
Item 9	Yes	No	Yes	Yes	No	No	Yes	No	N/A *	Yes	Yes	Yes	No	Yes	N/A *	No	Yes
Item 10	Yes	Yes	Yes	No	No	Yes	Yes	No	N/A *	Yes	No	No	No	Yes	N/A *	No	No
Item 11	Yes	Yes	No	Yes	Yes	Yes	Yes	Yes	Yes	No	No	No	Yes	No	Yes	No	Yes
Item 12	Yes	Yes	Yes	Yes	Yes	Yes	Yes	Yes	Yes	Yes	Yes	yes	Yes	Yes	Yes	Yes	Yes
Item 13	Yes	Yes	Yes	Yes	No	Yes	No	No	N/A *	Yes	No	No	No	Yes	N/A *	No	No
Item 14	Yes	Yes	Yes	Yes	Yes	Yes	Yes	Yes	Yes	Yes	Yes	Yes	Yes	Yes	Yes	Yes	Yes
Item 15	Yes	Yes	Yes	Yes	Yes	Yes	Yes	Yes	Yes	Yes	Yes	Yes	Yes	Yes	Yes	Yes	Yes
Item 16	Yes	Yes	No	Yes	Yes	No	No	No	Yes	No	Yes	No	No	Yes	No	No	No
Item 17	Yes	Yes	Yes	Yes	Yes	Yes	Yes	Yes	Yes	Yes	Yes	Yes	Yes	Yes	Yes	Yes	Yes
Item 18	Yes	Yes	Yes	Yes	Yes	Yes	Yes	Yes	Yes	Yes	Yes	Yes	Yes	Yes	Yes	Yes	Yes
Item 19	Yes	Yes	Yes	Yes	Yes	Yes	Yes	No	Yes	Yes	Yes	No	Yes	Yes	Yes	Yes	Yes
Item 20	Yes	Yes	Yes	Yes	Yes	Yes	Yes	Yes	Yes	Yes	Yes	Yes	Yes	Yes	Yes	Yes	Yes
Score	18	17	17	17	14	15	17	14	14 *	17	14	12	15	17	14 *	12	16

* This was a single-arm trial, and thus, there could be no randomization, concealment of randomization, or blinding of outcomes to investigators.

## Data Availability

Data and code utilized for data preparation and analyses are available on the Open Science Framework page for this project https://osf.io/5fjq3/.

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
