# Peer review of "Machine-Based Resistance Training Improves Functional Capacity in Older Adults: A Systematic Review and Meta-Analysis"

_jfmk, 2024, doi:10.3390/jfmk9040239_

Round 1
Reviewer 1 Report
Comments and Suggestions for Authors
The study appears to be understandable and provides a valuable contribution to the topic; however, I have identified some critical issues that I would like to share with the authors.
Abstract
Although the paragraph is labeled “Background/Objectives,” the objectives are actually presented in the “Methods” section.
In the background, it’s mentioned that strength training can be challenging (I would specify here that this applies to older adults, Line 11).
If the focus is on the fact that machine-based training is easier for older adults to perform, I would specify this in the objectives.
Line 16: I would suggest rephrasing the screening process and the distribution of studies into the two subgroups, "functional outcomes" and "strength outcomes".
Introduction
The benefits of using machines have been well reported; however, there are limitations, such as maintaining balance components and performing functional exercises. Bodyweight exercises represent a risk factor (as has been clearly stated), yet they also enhance components beneficial for maintaining cognitive, balance, and functional abilities, especially in the population under study. It might be worthwhile to add a couple of lines describing the importance of machine training as a valuable tool to support a program that takes every aspect, including functional aspects, into account.
2.1 Inclusion Criteria
Why was the 6-minute Walking Test not considered? It was surely used in many studies.
Line 135: How many studies included images only without reporting data? If there weren’t many, it could have been considered an exclusion criterion.
Table 3: In SMART-LD, the items should total 27, but only 20 are shown here. Did you intentionally decide not to consider certain items? If so, it would be better to include this in section 3.2.
Line 253: “Of the 17 studies, 6 used pneumatic resistance machines, one used a plate-loaded resistance machine, 7 studies used selectorized resistance machines, and 4 studies did not clarify resistance type, manufacturer, or model (though they did specify the use of machine-based resistance).”
The total number of studies reported adds up to 18 (6+1+7+4), not 17 as it should.
Line 357: a minor typo [61}
3.5 Study Duration and Frequency
Line 268: There is an inconsistency between the 18 studies and the 17 studies mentioned in the previous paragraph.
Line 291: How did you handle studies that did not report the workload?
Figures
I would avoid creating a single large figure that describes the forest plot, Global grand mean, and contrast between conditions, as this could make it inconvenient to refer to individual parts of the image (e.g., lower right panel, lower left panel). Perhaps it would be more intuitive to have Figure 1, Figure 2… Figure 6… or, alternatively using different letters for different images
Line 381: The reasoning here may be understandable, but the statement about acquiring strength should be backed up by some studies on skill transfer. It would be advisable to cite some studies about it.
Author Response
|
Reviewers’ comments |
Authors’ response |
|
Reviewer 1 |
|
|
The study appears to be understandable and provides a valuable contribution to the topic; however, I have identified some critical issues that I would like to share with the authors. |
We appreciate your time reviewing the manuscript and comments to help improve the quality of this publication. |
|
Abstract |
|
|
Although the paragraph is labeled “Background/Objectives,” the objectives are actually presented in the “Methods” section. |
The sentence beginning “The aims of this systematic review…” has been moved unto the Background/Objectives section of the Abstract |
|
In the background, it’s mentioned that strength training can be challenging (I would specify here that this applies to older adults, Line 11). |
Agreed, this has been added. |
|
If the focus is on the fact that machine-based training is easier for older adults to perform, I would specify this in the objectives. |
Agreed, this has been added in to the objectives section. |
|
Line 16: I would suggest rephrasing the screening process and the distribution of studies into the two subgroups, "functional outcomes" and "strength outcomes". |
The subtract reads: “Following screening 17, articles met inclusion criteria for the systematic review, 15 of which were included in the meta-analysis for functional outcomes (n=614 participants), and 11 of which were included in the meta-analysis for strength outcomes”. It is our opinion this is fitting with your comment; the subgrouping based on outcomes. |
|
Introduction |
|
|
The benefits of using machines have been well reported; however, there are limitations, such as maintaining balance components and performing functional exercises. Bodyweight exercises represent a risk factor (as has been clearly stated), yet they also enhance components beneficial for maintaining cognitive, balance, and functional abilities, especially in the population under study. It might be worthwhile to add a couple of lines describing the importance of machine training as a valuable tool to support a program that takes every aspect, including functional aspects, into account. |
We have added a statement within the introduction “However, we should clarify that the use of resistance machines and free-weights are not mutually exclusive; training programs might efficaciously choose to incorporate both resistance methods.” |
|
2.1 Inclusion Criteria |
|
|
Why was the 6-minute Walking Test not considered? It was surely used in many studies. |
We do recognise there are other tests of functional capacity, however, we felt that these were well represented in the literature and correlate well to balance, coordination and strength tasks. |
|
Line 135: How many studies included images only without reporting data? If there weren’t many, it could have been considered an exclusion criterion. |
As I recall there were only a couple of studies which included figures where data needed to be extracted, but we have comfortably done this in the past and have confidence in the tools. It seems a shame to exclude studies which do not include raw data (or adding that as a criterion) when there are already so few (n=17). |
|
Table 3: In SMART-LD, the items should total 27, but only 20 are shown here. Did you intentionally decide not to consider certain items? If so, it would be better to include this in section 3.2. |
The SMART-LD is a 20-item scale. We appreciate your comment, and we checked this. Perhaps you’re thinking of a different scale. This is quite new and specific to resistance training studies (https://osf.io/preprints/osf/nhva2). |
|
Line 253: “Of the 17 studies, 6 used pneumatic resistance machines, one used a plate-loaded resistance machine, 7 studies used selectorized resistance machines, and 4 studies did not clarify resistance type, manufacturer, or model (though they did specify the use of machine-based resistance).” The total number of studies reported adds up to 18 (6+1+7+4), not 17 as it should. |
Apologies, that this was not clear. We’ve now added a line “or any combination”. For example some studies used both plate loaded and selectorized, or selectorized and pneumatic. |
|
Line 357: a minor typo [61} |
Thank you for catching that! |
|
3.5 Study Duration and Frequency |
|
|
Line 268: There is an inconsistency between the 18 studies and the 17 studies mentioned in the previous paragraph. |
Yes, there were 17 studies, this has been amended. |
|
Line 291: How did you handle studies that did not report the workload? |
We did not incorporate load, repetition duration, volume, etc. in to the statistical analyses. Our discussion here is simply the systematic review where we are describing the studies. While some studies did not report some of this data (e.g., load, repetition duration, etc.) they typically reported other data (e.g., repetitions and effort) which could be used to estimate the training load. It’s our opinion that this is simply representative of the way many people program a workout, but doesn’t require a more detailed discussion in the manuscript. |
|
Figures |
|
|
I would avoid creating a single large figure that describes the forest plot, Global grand mean, and contrast between conditions, as this could make it inconvenient to refer to individual parts of the image (e.g., lower right panel, lower left panel). Perhaps it would be more intuitive to have Figure 1, Figure 2… Figure 6… or, alternatively using different letters for different images |
I appreciate the comment, however, we don’t make reference to specific parts of each figure. Further we don’t think it’s necessary to divide what can be a single figure in to multiple figures (each figure would be split in to 3 further figures). |
|
Line 381: The reasoning here may be understandable, but the statement about acquiring strength should be backed up by some studies on skill transfer. It would be advisable to cite some studies about it. |
A comment has been added here guiding readers to a previous publication which discusses strength training and specificity of movement, notably the negative implications of adding resistance to try to replicate sporting movements. |
|
Reviewer 2 |
|
|
The systematic review and meta-analysis presented to me for evaluation proved to be an interesting read. Its aim was to explore the impact of machine-based resistance training upon strength and functional capacity in older adults. In addition, the authors reviewed the variables characterising the training programme and their relationship to the outcome variables. |
|
|
Out of my duty as a reviewer, I have tried to catch the weaknesses of the text and to identify passages that require significant correction. It turned out that I had little to add. The article is well written and the argument logically flowing. My only substantive comment relates to the issue of safety during exercises. In the introduction, a lot of space was devoted to the benefits and positive effects of weight training and the safety issue was omitted (except one sentence in line56). The use of resistance exercise machines is particularly recommended for people who are new to training and are unfamiliar with the use of free weights but there are also some risks. An additional paragraph on an advantage of resistance machines would have been useful, as would a few sentences on the careful selection of exercise volume and intensity for older people. |
Some detail has been added which elaborates on the disparity between free-weights and resistance machines for safety issues and injury risk. |
|
I am satisfied with the statistical methods used for meta-analysis as well as selection of the references. |
Thank you. |
|
I must mention the lack of pre-registration of the review and meta-analysis, which the authors put in the limitations of the study. This is basically information for the edit6or, as I do not see it as something that undermines the value of the article. |
Thank you. It is apparently necessary to retrospectively register the article which has been done on osf: https://osf.io/e4586 |
|
I would not include in the text (e.g. line 136) who performed each activity - the contributions of the individual authors are included at the end of the article. |
This sentence has been removed. |
|
The description of the abbreviations and symbols used in Table 1 should rather be at the beginning of the table after the table header. The table is huge and for reasons of reading order, this would be more convenient. |
This has been moved to under the table header as suggested. |
|
I know that IQR is an interquartile range, but some readers may need clarification (line 269) |
This has been added in full prior to the first abbreviation |
|
In summary: I find the text valuable and clearly written. |
Thank you for your time and detailed comments. |
Reviewer 2 Report
Comments and Suggestions for Authors
The systematic review and meta-analysis presented to me for evaluation proved to be an interesting read. Its aim was to explore the impact of machine-based resistance training upon strength and functional capacity in older adults. In addition, the authors reviewed the variables characterising the training programme and their relationship to the outcome variables.
Out of my duty as a reviewer, I have tried to catch the weaknesses of the text and to identify passages that require significant correction. It turned out that I had little to add. The article is well written and the argument logically flowing. My only substantive comment relates to the issue of safety during exercises. In the introduction, a lot of space was devoted to the benefits and positive effects of weight training and the safety issue was omitted (except one sentence in line56). The use of resistance exercise machines is particularly recommended for people who are new to training and are unfamiliar with the use of free weights but there are also some risks. An additional paragraph on an advantage of resistance machines would have been useful, as would a few sentences on the careful selection of exercise volume and intensity for older people.
I am satisfied with the statistical methods used for meta-analysis as well as selectio0n of the references.
I must mention the lack of pre-registration of the review and meta-analysis, which the authors put in the limitations of the study. This is basically information for the edit6or, as I do not see it as something that undermines the value of the article.
Specific comments (suggestions rather than demands)
I would not include in the text (e.g. line 136) who performed each activity - the contributions of the individual authors are included at the end of the article.
The description of the abbreviations and symbols used in Table 1 should rather be at the beginning of the table after the table header. The table is huge and for reasons of reading order, this would be more convenient.
I know that IQR is an interquartile range, but some readers may need clarification (line 269)
In summary: I find the text valuable and clearly written.
Author Response

(The authors gave the same response as above.)

Round 2
Reviewer 1 Report
Comments and Suggestions for Authors
I would like to thank you for thoughtfully considering the comments and suggestions provided in the previous review. I appreciated the way you integrated the changes.
From my perspective, the suggested modification could have provided a bit more visual clarity, but I understand your decision to keep the figure unchanged and respect the rationale behind it. That said, the current figure is still effective and well executed, and it does not hinder the understanding of the content.
Line 11: Now it's more coherent
Line 268: Ok, now it is clear
line 395: This added comment reinforces the concept.